# Study protocol: The effect of a Medication Coordinator on the quality of patients' medication treatment (MEDCOOR)—Randomized controlled trial

**Maja Schlünsen**[1,2]*, **Trine Graabæk**[3], **Andreas Kristian Pedersen**[4], **Jan Dominik Kampmann**[2,5], **Lene Juel Kjeldsen**[1,2]

1 The Hospital Pharmacy Research Unit, University Hospital of Southern Denmark, Aabenraa, Denmark, 2 The Department of Regional Health Research, University of Southern Denmark, Odense, Denmark, 3 Hospital Pharmacy Funen, Odense University Hospital, Odense, Denmark, 4 Department of Clinical Research, University Hospital of Southern Denmark, Aabenraa, Denmark, 5 Department of Internal Medicine, University Hospital of Southern Denmark, Sønderborg, Denmark

* Maja.Schlunsen@rsyd.dk

**Data Availability Statement:** No datasets were generated or analysed during the current study. All

## Abstract

Patients' safety can be compromised in the transition of care between healthcare sectors. Optimal information flow across healthcare sectors and individualized medication treatment tailored to each patient is vital to prevent adverse events like drug-related problems. When medication changes are made during hospitalization, it is essential to ensure that the relevant general practitioner (GP) is included in the communication chain. This randomized controlled trial examines the effect of a Medication Coordinator who facilitates medication reviews in close collaboration with patients using *My Medication Plan*. Patients in the intervention group receive the medication review in combination with including suggested medication amendments documented in their electronic discharge letter send, which is sent to their GP. The patients randomized to the control group receive standard care by the ward staff. Seventy patients from the Endocrinology and Nephrology Unit at the Hospital Sønderjylland will be included in the intervention and control groups, respectively. The primary outcome is the proportion of potentially inappropriate medications. Secondary outcomes include patient-reported outcomes, i.e., quality of life and medication burden. Additional outcomes include the patient's medication risk score, whether the patient is readmitted, and whether the patient has contacted the staff at the hospital unit after the hospital discharge. The framework for complex intervention is applied, because it allows flexibility and adaption in meeting patients' needs by implementing tailored, possibly complex interventions in different healthcare settings. This project will examine a particular piece in the puzzle of the complexity of conducting a medication review and communication of suggested medication amendments to the patients, healthcare at the hospital, and the GP. Hopefully, this can contribute to a reduction in the risk of potentially inappropriate post-hospital medication usage.

**Trial registration:** The study has been registered at ClinicalTrial.gov with the registration number: NCT06383364. https://clinicaltrials.gov/study/NCT06383364.

relevant data from this study will be made available upon study completion.

**Funding:** (1) The University of Southern Denmark, one year salory to MS on 425.000 DKK and a grant of 10.000 DKK to be used on running cost. https://www.sdu.dk/en/forskning/phd/phd_skoler/phdskolensundhedsvidenskab (2) The Region of Southrn Denmark gave one year salory to MS on 436.000 DKK https://regionsyddanmark.dk/fagfolk/forskning/region-syddanmarks-forskningspuljer/ph-d-puljen (3) The University Hospital of Southern Denmark gave one year salory for MS on 572.000 DKK and a grant on 13.000 DKK to be used in running costs. https://sygehussonderjylland.dk/.

**Competing interests:** The authors have declared that no competing interests exist.

# 1. Introduction

Patient safety can be compromised in the transition between healthcare sectors, such as admission to the hospital, discharge from the hospital, visits to outpatients' clinics, and consultations with a general practitioner (GP) [1]. Approximately 20% of patients have experienced at least one adverse event in the transition between healthcare sectors [2]. Information flow across healthcare sectors must be optimized, and medication treatment tailored to each patient to prevent adverse events [1,3,4]. This type of research can be accommodated using the framework for complex interventions described by the Medical Research Council (MRC) [5]. Complex interventions allow flexibility and adaptation to individual patients' needs across healthcare settings during implementation [5]. The outcomes of complex interventions are due to the overall intervention rather than one specific component [5].

Patients and relatives express needs for more information about their medication treatment when transitioning from hospital to home [6–9]. A consequence of discontinuity of medication information might be adverse events like drug-related problems or unintentional non-adherence to medication [7,10–12] with aggravation of the patient's health and illness [11–13].

Electronic medication records can be implemented to ensure access to updated medication lists across healthcare sectors [4–16]. The electronic Shared Medication Record (SMR) is used in Denmark, and the SMR comprises all prescriptions and purchases of medication from the community pharmacies [14]. Based on the patient's medication purchases, it is possible to assess patient's adherence to medication. Despite the intention to ensure an accurate and updated medication list, discrepancies between electronic records and the patient's actual medication use have been demonstrated [17–20]. Medication discrepancies can be due to patient non-adherence or changes to the medication regimen after medication purchase, including discontinuation of medication treatment, incorrect dosage and timing of the dosage, and omission of a drug [17,18]. This could result in drug-related problems posing a risk to patient safety [21].

Potentially inappropriate medications (PIMs) can be used in research to reflect the quality of older patients' medication treatment [22,23]. PIMs constitute a risk for causing drug-related problems such as falls, hospitalizations, increased morbidity and mortality [24–31]. Screening tools such as *Screening Tool of Older Persons' Prescriptions* (STOPP) and *Screening Tool to Alert to Right Treatment version 3* [32] and *Medication Appropriateness Index* [33] have been developed to assess if patients are at risk of drug-related problems. These tools can also be used as means to describe the quality of medication treatment, but the usefulness may be inadequate [34]. In Denmark, an addition of the *Discontinuation List 2024/2025* (*Seponeringslisten 2024/2025*) [35], *The Seven High Risk-Situation Drugs* [36] and *Medication Risk Score* (MERIS) [31] have been developed as screening tools. However, none of these screening tools has been assessed regarding the prioritization of patients most in need of clinical pharmacy interventions like medication reviews. Due to scarce resources, there is a need to identify patients who will benefit the most from receiving a medication review. The screening tools can assist healthcare professionals when prioritizing patients who could benefit from a medication review with the purpose to optimize the patients medication treatment [31,37].

Medication reviews comprise a systematic and structured process where the patient's medication treatments are critically assessed [14,26,30]. Medication reviews aim to optimize the use of medicines and minimize drug-related problems. Positive effects have been reported on the quality of medication use and reduced hospital readmissions [38–41]. However, further information about patient-reported outcomes, such as quality of life and treatment burden, is needed [40].

Communication between healthcare professionals after transitioning between sectors is essential to ensure adherence to medication changes. In Denmark, digital platforms exist to support communication, e.g. discharge letters between sectors. However, the GPs need specific instructions about managing a patient's medication treatment after hospital discharge [12,42]. The absence of instructions could explain why medication changes made during hospitalization sometimes are not effectuated [43]. In such instances, a Medication Coordinator could be used to coordinate the communication between healthcare sectors.

A tool, *My Medication Plan*, was developed to support communication between sectors and empower patients to gain control over their medication treatment. The development of *My Medication Plan* is described in an unpublished study [conducted by Schlünsen M, Kjeldsen LJ, and Hansen GT. The development of *My Medication Plan* involving patient and patient representatives as co-designers, for submission]. *My Medication Plan* was created involving patients and patient representatives as co-designers to accommodate the patients' needs for information and empowerment. *My Medication Plan* combines the patients' updated SMR with information about appointments across different healthcare settings.

This randomized controlled trial aims to examine the effect of a Medication Coordinator on the quality of medication treatment measured by PIMs among hospitalized patients who use at least five medications. The Medication Coordinator facilitates medication reviews in close collaboration with the patients by applying motivational interview concepts in combination with *My Medication Plan*. The primary objective is determining the proportion of potentially inappropriate medication after hospital discharge. Secondary objectives include synthesizing data on patient-reported outcomes on quality of life and treatment burdens measured through surveys. In addition, an assessment of hospital readmission and patient contact to the ward will be included.

## 2. Methods and analysis

### 2.1. Study design

The study is a single-center randomized controlled trial assessing a complex intervention with the Medication Coordinator. The screening, enrolment, randomization to the interventions, and assessments will be conducted by MS only (Fig 1). The reporting of this protocol follows the guideline for Standard Protocol Items: Recommendation for Interventional Trials–SPIRIT 2013 [44] (S1 File). The randomized clinical trial reporting will follow the CONSORT guidelines [45].

**2.1.1. Recruitment of patients.** Patients will be recruited from the Endocrinology and Nephrology Unit at the Hospital Sønderjylland. The ward treats renal disease, diabetes, hormone disorders, patients with diabetic ulcers, and internal medicine patients. The ward comprises 16 beds distributed by rooms for two or four patients. Recruitment will commence in *medio* May 2024 and be completed by *ultimo* May 2025. We expect to include four patients weekly.

The inclusion criteria comprise all hospitalized patients 18 years or above, and are prescribed at least five medications specified in the Electronic Patient Journal (EPJ).

Patients are not eligible for inclusion if they are unable to communicate in Danish, are cognitively impaired, e.g. suffering from dementia or Alzheimer's, or cannot cooperate due to, e.g. hallucination or aggressive behavior.

The Medication Coordinator will screen patients for eligibility and include relevant patients in the study. After the initial screening, the ward nurses are consulted regarding patient capability to contribute to a conversation about medication based on the patients habitual

|  | STUDY PERIOD | | | | | | | |
|---|---|---|---|---|---|---|---|---|
|  | Enrolment | Allocation | Post-allocation | | | | | |
| **TIMEPOINT** | $-t_1$ | **0** | $t_1$ | $t_2$ | $t_3$ | $t_4$ | $t_5$ | $t_6$ |
| **ENROLMENT:** | | | | | | | | |
| *Eligibility screen* | X | | | | | | | |
| *Informed consent* | X | | | | | | | |
| *Allocation* | | X | | | | | | |
| **INTERVENTIONS:** | | | | | | | | |
| *Intervention Group* | | X | X | X | X | X | X | X |
| *Control Group* | | X | X | | | | X | X |
| **ASSESSMENTS:** | | | | | | | | |
| *Identification number* | X | | X | | | | | |
| *Age* | | | X | | | | | |
| *Sex* | | | X | | | | | |
| *Reason for admission* | | | X | | | | | |
| *Number of and ATC-codes of medication listed in SMR* | | | X | | | | | |
| *Primary outcome* | | | | | | | | |
| *Number and ATC of PIMs listed in SMR* | | | X | X | | | | X |
| *Secondary outcome* | | | | | | | | |
| *Medication Risk Score (MERIS)* | | | X | X | | | | |
| *Follow-up after hospital discharge* | | | | | X | | | |
| *Questionnaire, EQ-5D-5L with EQ VAS + QoL VAS, MTBQ + Treatment Burden VAS* | | | X | X | | | X | |
| *Readmissions after hospital discharge* | | | | | | X | | |
| *Contact to ward /Med.Coor after hospital discharge* | | | | | | X | | |

**Fig 1. SPIRIT schedule.** The schedule highlight the timeline for patient enrollment, plans for assessment of outcomes measures during the study period. Abbreviation: $t_1$ = baseline at inclusion, $t_2$ = at hospital discharge, $t_3$ = approximately 7 days after hospital discharge $t_4$ = 30 days after hospital discharge, $t_5$ = 3 months after hospital discharge, $t_6$ = 6 months after hospital discharge. Abbreviations: ATC = Anatomical Therapeutic Chemical Classification System, SMR = Shared Medication Record, PIMs = potentially inappropriate medications. VAS = Visual Analogue Scale, QoL = Quality of life, MTBQ = Multi-morbidity Treatment Burden Questionnaire, Med. Coor = Medication Coordinator.

condition. The Medication Coordinator performs this task so the everyday workload of the ward staff is not increased or disturbed.

## 2.2. Framework for complex interventions

The MRC framework for complex intervention describes four phases: 1. Develop or Identify intervention, 2. Feasibility, 3. Evaluation, and 4. Implementation [5]. It is important to reveal what works for whom (patients), how it works (components), and under which circumstances (settings) the complex intervention is most effective [46,47]. We have developed a program theory to visualize the different components of the complex intervention (Fig 2) [5,47,48].

The intervention in this study is dynamic and thus complex due to many different components, and the effect might rely on the patient's subjective perceptions.

**2.2.1. The intervention group.** The complex intervention was inspired by Ravn-Nielsen et al. [49] and included some adjustments. The Medication Coordinator facilitates the medication reviews in close collaboration with the patients by applying concepts of motivational interview in combination with *My Medication Plan* [50,51] [Unpublished study conducted by Schlünsen M, Kjeldsen LJ, and Graabæk T. The development of *My Medication Plan* involving patient and patient representatives as co-designers, for submission]. Furthermore, the

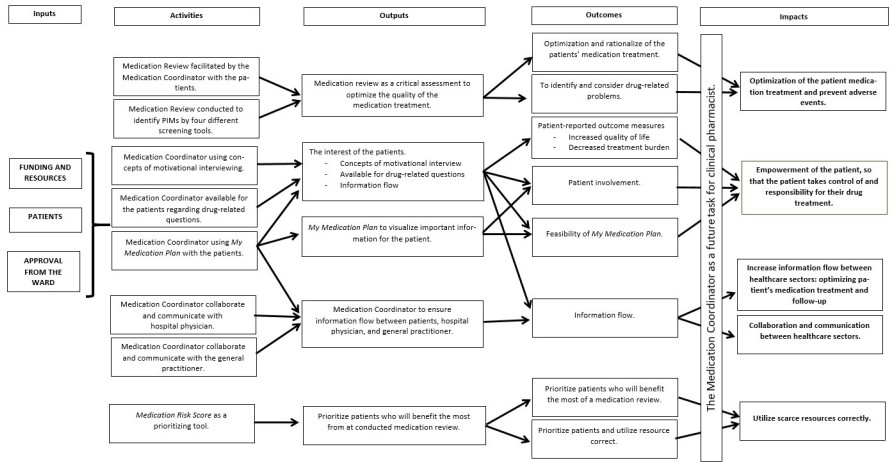

**Fig 2. Program theory for the study design.** The program theory describes the complex intervention with inputs, activities, outputs, outcomes, and impacts. PIMs = potentially inappropriate medication.

Medication Coordinator facilitates the flow of information about medication treatment in the transition of care between healthcare sectors (Fig 3).

Based on the inclusion criteria, the patient's contribution to the intervention and the intervention itself, the patient can withdraw his/her consent or die, otherwise there will be no criteria for the patient not being able to continue in the study after allocation.

**2.2.2. The control group.** The patients in the control group receive usual treatment from a team consisting of hospital physicians, nurses, nurse assistants, and if required, occupational therapists, physiotherapists, and clinical dieticians. Medication reconciliation might be a part of the patients' usual care performed by a pharmaconomist performed at the emergency department prior to allocation to another ward or a hospital physician at the ward. Hospital physicians and/or nurses might counsel patients about medication treatment during hospitalization (Fig 3).

**2.2.3 Medication coordinator.** The Medication Coordinator's tasks are as follows:

1. To empower the patients to gain control of their medication treatment [52].

2. To collaborate with the patients through established confidence.

3. To coordinate the communication of medication recommendations, medication changes, and implementations of the medication review made during hospitalization to the patients.

4. To assist the patient in noting medication changes made during hospitalization and follow-up appointments across sectors in *My Medication Plan*, if desired by the patients.

5. To contact the patients for follow-up about health status and any drug-related questions 7 days after hospital discharge by telephone.

**2.2.4. Medication review.** The medication reviews are conducted to optimize the patients' medication treatment. The process of optimization is through the identification of any PIMs that may be discontinued and to identify if the patients experience any drug-related problems that need to be considered. Hence, the medication review is expected to increase the quality of the patients' medication treatment.

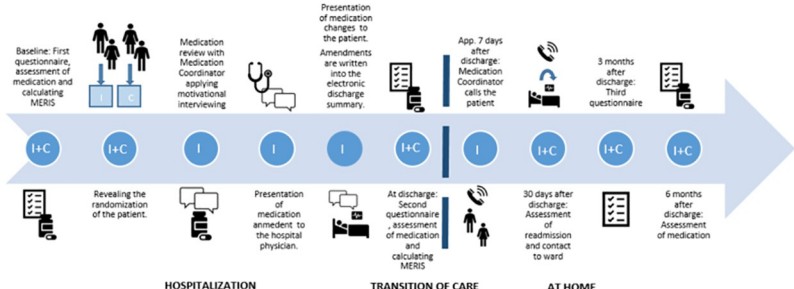

**Fig 3. Flow of the patient in the intervention group (I) and control group (C).** Primary and secondary outcome measures are collected at inclusion, discharge, and 30 days after discharge for the evaluation of readmission and six months after for the remaining outcome measures. The dashed line represents the transition of care. Medication Risk Score = MERIS.

The definition of prescribing quality indicator used is by Lawrence and Olesen [53]: "*a measurable element of prescribing performance for which there is evidence or consensus that it can be used to assess quality, and hence in changing the quality of care provided* [53]". Inspired by Wallerstedt et al [34] the measureable element of prescribing performance used to assess the quality of the patients' medication treatment is the number of PIMs. The patients' medications are registered according to the Anatomical Therapeutic Chemical (ATC) Classification System at level 5 [54]. The medication list is assessed for PIMs by applying: two national screening tools *(Discontinuation List 2024/2025* [35] and *the Seven High-risk Situation Medications* [36], and two international screening tools (STOPP [55] and MERIS [31,37].

The medication reviews are conducted as published by Graabæk et al. [56] but with an addition of a sixth, seventh and eight step to the existing five steps.

The patients are presented with the changes made to the medication treatment in the sixth step. The patients are encouraged to write down any important information in *My Medication Plan*. The suggested medication amendments are not implemented prior to hospital discharge, because these changes may require that the patient has been stabilized. Step 7 includes specific instructions about medication amendments in the electronic discharge summary forwarded to the GP.

As the eight step is a follow-up phone call to the patient approximately seven days after discharge. The follow-up conversations is based on the medication review carried out during hospitalization.

The concepts of motivational interviewing are applied during the medication review to focus on the patients' needs and wishes [50,51,57,58].

**2.2.5. My medication plan.** *My Medication Plan* is a non-electronic patient booklet consisting of a print of the SMR and specific documents sharing important medication-related information. *My Medication Plan* was developed to empower the patients and as a contribution to the information flow in the transition of care. *My Medication Plan* is available upon request albeit in Danish.

## 2.3. Data collection

Demographic variables, primary outcome and secondary outcomes will be collected at marked points (Fig 1). Some data will be collected through EPJ, an electronic healthcare system, utilized by the Region of Southern Denmark.

**2.3.1. Demographic variables.** Demographic variables collected at baseline will include the patient's age, sex, reason for hospital admission, and number of medications.

**2.3.2. Primary outcome.** The primary outcome is the proportion of PIMs after hospital discharge. The PIMs are defined by four different screening tools, hence if at least one tool indicates that a medication as a PIM, we will define the medication as a PIM [31,35,36,59].

**2.3.3. Secondary outcome.** The secondary outcomes are patient-reported outcomes about quality of life and treatment burden after 3 months, but also patient's readmissions, and contact to the ward after 30 days. The patient's MERIS score is calculated to estimate if a medication review could change the MERIS score and reduce the probability of experiencing a drug-related problem.

*2.3.3.1. Patient-reported outcome.* To measure patient-reported outcomes regarding the quality of life and treatment burden, the *EuroQol-5* Domain (EQ-5D) and Multi-morbidity Treatment Burden Questionnaire (MTBQ) are used [55,57]. The questionnaires are combined with a Visual Analogue Scale (VAS) regarding quality of life and treatment burden as validation. VAS assessment of subjective phenomena will be converted to numerical data [60].

The patients will fill out EQ-5D, EQ-VAS, MTBQ, and the two VAS's at inclusion and hospital discharge. The same questionnaires are answered three months after hospital discharge, either by e-mail or telephone, conducted by the first author.

EQ-5D contains five questions regarding five domains: mobility, self-care, usual activities, pain/discomfort, and anxiety/depression, each with five answer categories: no problems, slight problems, moderate problems, severe problems, and extreme problems. Permission to use the Danish EQ-5D version has been granted by EuroQol [61].

The EQ-VAS is a vertical VAS in which patients report their perceived quality of life from 0–100 [61]. The endpoints are labelled with "*The worst health you can imagine*" (0) and "*The best health you can imagine*" (100) [61]. These measurements are used to quantify a health outcome that reflects the patient's perception.

An additional VAS regarding quality of life is added to validate if the VAS can be applied instead of a questionnaire. The questionnaire used for the validation is the ED-Q5.

The VAS is vertical and ranges from zero (lowest quality of life) to 100 (highest quality of life) by numbers. The question asked is: "How is you quality of life today?" The endpoints are labelled with "*The worst quality of life you can imagine*" (0) and "*The highest quality of life you can imagine*" (100).

The MTBQ is a validated tool to assess patients' perceived treatment burden [62]. Treatment burden describes patients' perceptions of the effort required to look after their health and its effect on everyday life [63]. A Danish version of the MTBQ has been validated [64] and will be used with permission.

The MTBQ is a ten-item questionnaire with good content validity, high internal reliability, and good construct validity [62]. The questions cover medication management, self-monitoring, contact with healthcare professionals, obtaining information, implementing lifestyle changes, and relying on help [62,64]. The answers are designed with a five-point Likert scale with the possibilities: "*Not difficult*" (0), "*A little difficult*" (1), "*Quite difficult*" (2), "*Very difficult*" (3), "*Extremely difficult*" (4), and "*Does not apply*" (0) [62]. The MTBQ scores are categorized into no burden (score 0), low burden (score < 10), medium burden (score 10–22), and high burden (score ≥ 22) [62,64].

An additional VAS regarding treatment burden will be added to validate if the VAS can be applied instead of a questionnaire. The questionnaire used for the validation is the MBTQ.

The VAS is vertical and ranges from zero (no burden at all) to 100 (the highest burden imaginable) by numbers. The questions asked is: "How much of a burden is it for you to manage your illness today?" The endpoints are labelled with "*The highest burden it is for you to manage your illness*" (0) and "*The least burden it is for you to manage your illness*" (100).

*2.3.3.2. Hospital readmission.* Patient readmissions are assessed via EPJ 30 days after hospital discharge. As the data regarding readmissions will only be collected through EPJ—readmission data can only be collected from the Region of Southern Denmark [no other regions in Denmark].

*2.3.3.3. Contact to the ward.* Patient contact with the ward will be evaluated based on justification for the contact, e.g. medication-related questions. Data will be collected via EPJ 30 days after hospital discharge.

*2.3.3.4. Patients' Medication Risk Score.* The patient's MERIS will be calculated at inclusion and hospital discharge using the Software Robot Sirenia Manatee [31,37]. The MERIS algorithm provides a score reflecting the risk of probability for the patients experiencing a serious drug-related problem [37]. The MERIS score is calculated for all patients included in the study to assess which patients would benefit most from a medication review. The calculation is based on the patients' most recently measured, estimated glomerular filtration rate value, the number of medications, and the risk of a drug to cause harm or interactions. The medications with risk for harm or interactions are described in the literature [29,65,66]. Data will be collected via EPJ.

*2.3.3.5. Follow-up after discharge.* In the follow-up conversation between the patient and the Medication Coordinator, data about which medication(s) and which topic regarding the medications(s) are collected. Furthermore, data about the usage of *My Medication Plan* are collected to be used in the reporting of the feasibility of the complex intervention.

## 2.4. Sample size

To calculate power, we used a Monte Carlo simulation. We based the simulation on the beta-regression [67]. We assumed the proportion of PIMs were distributed across different time points, as highlighted in Table 1, similar to results reported by McCarthy et al. [68]. The Do-File from STATA/BE used for the calculation is found in S2 File.

The variance was set at 0.5, and we assumed the risk of dropout after 6 months was associated with age (see formula below)

$$P(dropout) \ = \ exp(1.1*age - 70)/(1.1*age - 70 + 1) \tag{1}$$

We calculated the p-value using the likelihood ratio test because we have multiple time points. Under the assumption that age will be related to the dropout rate, we needed 70 patients in each group to achieve a power of 93% at an alpha level of 0.05.

## 2.5. Randomization

One hundred and forty patients (70 in each group) will be included and randomized. A simple block randomization method will be used [69], with patients randomized by a computer-generated randomization sequence program using REDCap [70,71]. Variable blocks are used to adjust for assignment bias. The Medication Coordinator will be blinded to the size of the blocks and will not be able to predict patient group allocation.

**Table 1.** *Data used for the power calculation.*

| Time point | At hospital discharge | Six months after hospital discharge |
|---|---|---|
| Proportion of PIMs in intervention group | 15% | 13% |
| Proportion of PIMs in control group | 10% | 12% |

Data from McCarthy et al. [68] used to calculate the sample size. Abbreviation: PIMs = Potentially Inappropriate Medication

## 2.6. Statistics analysis

Descriptive statistics will be utilized to check for exchangeability between the two groups according to baseline variables. Non-categorical data will be presented as mean and standard deviation and analyzed using a t-test or median and inter-quartile ranges analyzed with Wilcoxon rank sum test, depending on whether the given variable follows a normal distribution. Depending on Cochran's rule, categorical data will be presented as numbers or proportions and analyzed with either a Fisher or Chi-square test. Survival outcomes will be presented with Kaplan-Meier curves.

To assess whether interaction terms will be needed in the primary analysis, we will construct a margins plot highlighting the mean of each group at each time point (baseline, at hospital discharge, and six months after hospital discharge).

The final manuscript will report 95% confidence intervals and two-sided p-values. A p-value below 5% will be considered statistically significant.

All data will be analyzed in STATA/BE version 18 or R.

The data analyst will be blinded for group allocation during the statistical analysis.

**2.6.1. Primary analysis.**   The primary analysis will be a beta-regression with clustered standard errors as we have repeated measurements, thus adopting a population-averaged approach. We will incorporate an interaction between group and time, as we are unsure if the treatment is stable over time. Therefore, we will present the estimates for each time point (baseline, at hospital discharge, and six months after hospital discharge) and construct a margins plot based on the model. We will conduct a global test to test whether our intervention affects the primary outcome across all time points. If the test is statistically significant, we will conduct a z-test at each time point to assess when there is a statistically significant treatment effect. To avoid inflating the type I error rate in these z-test, we will adjust the z-test for multiple testing.

The primary analysis will follow the intention-to-treat principle; hence, missing data will need to be managed. As the beta regression uses maximum likelihood estimation, we can assume that the beta regression will yield unbiased results if the data is missing at random or completely at random. However, under the assumption of missing not at random, the estimates from the beta regression will be biased therefore, we will conduct joined modelling to assess the influence of missing data if the data was missing not at random. In the joint model, we will use logistic regression to estimate the risk of dropout.

**2.6.2. Analysis of secondary outcome.**   As quality of life, treatment burden, and patients' medication risk score all follow a Likert scale, we will use beta-binomial regression. The beta-binomial assumes our outcome follows a binomial distribution where the probability of landing in a given category follows a beta distribution to account for over-dispersion. As both outcomes are measured at different time points and the beta-binomial regression uses a maximum likelihood estimation, we will adopt the same test and manage missing data similarly to the primary analysis. In order to get interpretable effect sizes, we will use G-computation, where the Q-model will be a linear regression.

Our follow-up time is 30 days, so we will conduct a binomial regression to calculate the risk difference. Furthermore, we will view the above-mentioned secondary outcomes as time-to-event outcomes. Hence, we will use the Cox-proportional hazard model to ensure that our results are not dependent on the length of our study. The proportional hazard assumption will be graphically assessed using scaled Schönfeld residuals. Binomial regression is the primary analysis applied, while Cox regression is a sensitivity analysis.

## 2.7. Data management

The data collected will be stored in accordance with the Danish Law about General Data Protection Regulation (GDPR). All data will be stored and processed with the online-based

Research ElectronicDataCapture system (REDCap) (version 13.7.18) [70] through Odense Patient Data Explorative Network (OPEN) [72]. The data storage via OPEN and REDCap is compatible with the Danish Code of Conduct for Research Integrity [73]. REDCap databases are used to register all data about patient demographics, medication lists, randomization, and answers from the three questionnaire. Only individuals with permission will be allowed to access the patient databases.

This study will share data in accordance with the FAIR principles [74]. Data obtained about medication can be accessed through Danish Registers and, therefore, will not be made public to ensure anonymity and confidentiality. The Do-files from STATA/BE version 18 and R scripts made during the statistical analysis will be shared as Metadata.

## 2.8. Ethics

The study was registered with Clinical Trials with the trial number NCT06383364 [75]. Amendments to the protocol will be communicated to ClinicalTrials.gov.

No ethics approval was needed for this study, according to the Region of Southern Denmark's Scientific Ethics Committee, case number: 20232000–84. However, good ethical standard will be adhered to [73,76].

The study is registered in the Region Directory, case number 23/54272.

Patients included in the study will give written informed consent according to the Declaration of Helsinki [77] with additional permission to publish data and results anonymously in scientific articles and to other relevant stakeholders.

## 3. Discussion

This study is a randomized controlled trial, which will test the effect of a Medication Coordinator in the transition of care between healthcare sectors. The Medication Coordinator will facilitate the medication reviews in close collaboration with patients applying motivational interview concepts in combination with *My Medication Plan*. A strength of this study is that the intervention is complex. The MRC framework allows flexibility and adaptability during the process of performing the complex intervention [5]. The complex intervention will be adapted based on the patient's needs. Some patients want full involvement in every decision and aspect regarding their medication treatment. Meanwhile, other patients do not, but indeed wish to be informed about medication changes [78–80].

A primary outcome is selected to assess the overall effect of the intervention model. In the literature, several primary outcomes are used to report the effectiveness of medication reviews, and the effectiveness of the medication reviews are in consistent [39–41,81–83]. Primary outcomes often measured in randomized controlled trials when the intervention is a mediation review include mortality, acute readmissions, and PIMs [39,41,82]. These outcomes relate to patient safety and the optimization of a patient's medication treatment. The performance and effects of a medication review may differ across healthcare settings and clinical pharmacist qualifications. Due to the complexity of a medication review, the effect may have many components [5] and might not be able to detect by one primary outcome The limitation of this complex intervention is the inability to assess which part of the intervention provided the effect or was most effective. It is impossible to identify one component of a complex intervention and implement only this. However, we can document the implementation of components by process measures.

The effect of a medication review not only depends on the different components of the intervention itself but is also highly dependent on the patient's subjective experience [5,47]. Thus, the complex intervention model may have several positive effects, which we will assess

using secondary outcomes like patient-reported quality of life and treatment burden. Some patients might find the medication review with the Medication Coordinator effective, but for others, the information passed onto the GP in the discharge summary may be more effective. Nevertheless, all components are a part of the complex intervention.

Because the resources in the healthcare sector are scarce, prioritizing the patients most in need of a medication review is a priority [84]. A Danish study in general practices has demonstrated that patients with a MERIS score above 13 would most likely benefit from a medication review [37]. The MERIS screening tool is based on patients' renal function, the total number of medications, and the risk of harm or interactions caused by drugs [31,37]. These factors might be just as high a risk for experiencing an adverse drug-event as age. Age has been associated with increased risk of experiencing an adverse drug-event due to changes in pharmacokinetic and pharmacodynamic [85]. This may be explained by many drugs being renal excreted, hence may need dose-adjustments. These factors might indicate a risk for experiencing an adverse drug-event similar to age, which has been associated with increased risk of experiencing an adverse drug-event. This may be explained by many drugs being excreted renally, hence may need dose-adjustments. Therefore, there is a need to adjust the dose according to renal function. Patients of all ages are included in this study as younger patients could also have a high MERIS score and benefit from the intervention.

Another strength of this study is the intention to secure the information flow in the transition from the hospital to the patient's home. Medication changes and follow-up appointments with healthcare professionals are presented to the patient by *My Medication Plan* and to the GPs via the electronic discharge summary.

A strength of the data generation process is that one person collects it. It can also be a limitation of the study as there is a risk of the Pygmalion Effect [86]. It is a limitation that the first author is performing the complex intervention.

Patients in both groups are included from the same ward, which might result in contamination bias [87]. The physicians treat patients from both groups and will likely adopt some medication-related learning. The physicians may apply this learning to some of the patients in the control group, resulting in an increased focus on the patient's medication treatment. These limitations would bias the estimation of the effect towards no difference in relation to the null hypothesis.

Three questionnaires are to be answered by the patients, so there is a risk of transfer bias [88]. Patients might be particularly susceptible to loss of follow-up in Questionnaire 3, e.g. because they forget to answer or could pass away in the follow-up period. To avoid the loss of data, an intention-to-treat analysis is applied [89].

There is a risk for social-desirability bias in relation to the questionnaire [90,91], e.g., because the patients are answering what they think is expected of them. The same bias might occur during the medication review with the patients telling the Medication Coordinator what they think the Medication Coordinator requests to hear.

As hospitalized patients are often vulnerable, it can be difficult for them to answer a questionnaire. Especially if the questionnaires are long, they can be cognitively demanding. For this reason, the addition of the two VAS's will capture patient-reported outcomes regarding quality of life and treatment burden.

### 3.1. Practical implications

This project may have the capability to provide a missing piece to a puzzle, i.e., the inclusion of GPs and *My Medication Plan* in this collaborative model. The Medication Coordinator could

be a clinical pharmacist employed at a hospital or general practice. Hopefully, this can contribute to a reduction in the risk of post-hospital potentially inappropriate medication usage.

Patients in the intervention group are urged to become empowered and to take control over their medication treatment, as they are a part of the whole process that may increase their quality of life and reduce their treatment burden. Hence, the patient's medication treatment quality is optimized, reducing the risk of drug-related problems.

## Supporting information

**S1 File. SPIRIT checklist.**
(PDF)

**S2 File. Do File for the power calculation.** The do file used for the power calculation.
(PDF)

**S3 File.**
(PDF)

**S4 File.**
(PDF)

## Acknowledgments

Data management was provided and REDCap was hosted by OPEN, OPEN, Open Patient data Explorative Network, Odense University Hospital, Region of Southern Denmark [https://open.rsyd.dk/] [OPEN—Open Patient data Explorative Network]].

## Author Contributions

**Conceptualization:** Maja Schlünsen, Trine Graabæk, Lene Juel Kjeldsen.

**Formal analysis:** Maja Schlünsen, Andreas Kristian Pedersen.

**Funding acquisition:** Maja Schlünsen.

**Methodology:** Maja Schlünsen, Trine Graabæk, Andreas Kristian Pedersen, Jan Dominik Kampmann, Lene Juel Kjeldsen.

**Project administration:** Maja Schlünsen, Lene Juel Kjeldsen.

**Software:** Maja Schlünsen, Trine Graabæk, Lene Juel Kjeldsen.

**Supervision:** Maja Schlünsen, Lene Juel Kjeldsen.

**Writing – original draft:** Maja Schlünsen.

**Writing – review & editing:** Maja Schlünsen, Trine Graabæk, Andreas Kristian Pedersen, Jan Dominik Kampmann, Lene Juel Kjeldsen.

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
