## [Decision Letter · Decision Letter 0]

3 Jul 2024

PONE-D-24-18485Study Protocol: The Effect of a Medication Coordinator on the Quality of Patients’ Medication Treatment (MEDCOOR) – Randomized Controlled TrialPLOS ONE

Dear Dr. Schlunsen,

Thank you for submitting your manuscript to PLOS ONE. After careful consideration, we feel that it has merit but does not fully meet PLOS ONE’s publication criteria as it currently stands. Therefore, we invite you to submit a revised version of the manuscript that addresses the points raised during the review process.

We look forward to receiving your revised manuscript.

Kind regards,

Ramune Jacobsen

Academic Editor

PLOS ONE

 [(1) The University of Southern Denmark, one year salory to MS on 425.000 DKK and a grant of 10.000 DKK to be used on running cost. 

https://www.sdu.dk/en/forskning/phd/phd_skoler/phdskolensundhedsvidenskab

(2) The Region of Southrn Denmark gave one year salory to MS on 436.000 DKK 

https://regionsyddanmark.dk/fagfolk/forskning/region-syddanmarks-forskningspuljer/ph-d-puljen

(3) The University Hospital of Southern Denmark gave one year salory for MS on 572.000 DKK and a grant on 13.000 DKK to be used in running costs.  

https://sygehussonderjylland.dk/].  

5. Ethics statement only appears at the end of the manuscript:

Your ethics statement should only appear in the Methods section of your manuscript. If your ethics statement is written in any section besides the Methods, please move it to the Methods section and delete it from any other section. Please ensure that your ethics statement is included in your manuscript, as the ethics statement entered into the online submission form will not be published alongside your manuscript. 

Reviewers' comments:

Reviewer's Responses to Questions

**Comments to the Author**

1. Does the manuscript provide a valid rationale for the proposed study, with clearly identified and justified research questions?

Reviewer #1: No

Reviewer #2: Yes

Reviewer #3: Partly

2. Is the protocol technically sound and planned in a manner that will lead to a meaningful outcome and allow testing the stated hypotheses?

Reviewer #1: No

Reviewer #2: Yes

Reviewer #3: Partly

3. Is the methodology feasible and described in sufficient detail to allow the work to be replicable?

Reviewer #1: Yes

Reviewer #2: Yes

Reviewer #3: No

4. Have the authors described where all data underlying the findings will be made available when the study is complete?

Reviewer #1: No

Reviewer #2: Yes

Reviewer #3: Yes

5. Is the manuscript presented in an intelligible fashion and written in standard English?

Reviewer #1: No

Reviewer #2: Yes

Reviewer #3: Yes

6. Review Comments to the Author

You may also provide optional suggestions and comments to authors that they might find helpful in planning their study.

Reviewer #1: The authors designed a Randomized Controlled Trial to measure the Effect of a Medication Coordinator on the Quality of Patients’ Medication Treatment (MEDCOOR). However, the structure of the protocol does not fit with the guideline of the journal. Several breaks between paragraphs and unnecessary underlines hinder the quality of the paper. The authors did not define or operationalize “Quality of Patients’ Medication Treatment” . Therefore, the topic and the objective that is going to be measured are not parallel. I would suggest a careful review of the protocol and its feasibility before implementation. Publication of the protocol in its current form is barely considered.

You might consider the following points to improve your work.

No Please reduce the number of keywords.

Lines 33-34“communication of communicating suggested medication amendments to the patients” should be revised.

Line 190-192, Please describe the tool.

Line 203-205, no tool validation information is included.

Reviewer #2: Dear Authors

Thank you for your submission.

The comments will be available through journal editor.

Regards

Reviewer #3: The authors describe a protocol of a study which aims to evaluate the effect of a coordinating intervention on the quality of medication treatment. The primary objective of the planned study is the proportion of potentially inappropriate medication (PIM) six months after hospital discharge.

The description is detailed in many places, other points required by the SPIRIT guideline should be further elaborated

Item 14; The sample size calculation in the manuscript is based on different assumptions and lead to a different number of patients to be includes (e.g. 93% power in the manuscript vs. 80% power in the study protocol resulting in n=70 patients per group in the manuscript or N=80 in the study protocol). Please explain deviations between the documents.

The sample size calculation based on the assumption that the expected proportions of PIMS in the intervention group is higher as in the control group. Please explain why this assumption is reasonable.

Item 15; Please add to the description of the clinic's resources which strategies will be used for achieving adequate participant enrolment to reach target sample size.

Item 16a of the SPIRIT checklist suggests keeping details of any planned restriction (eg, blocking) in a separate document to conceal this knowledge to those who enrol participants or assign interventions. So the information from line 273 ff “A simple block randomization method …” should be transferred to a statistical analysis plan.

Item 16b; Not knowing the block length alone cannot avoid predictability. Further measures to enable concealment should be considered and described in the protocol.

Item 16c; Details of the Implementation of the allocation list should be provided. Who will generate the randomization list with REDCap, who will enrol patients and who will assign participants to interventions?

Item 20a; In general, the primary hypothesis, the sample size calculation and the statistical analysis should follow the same statistical approach. That means, if the primary objective is the comparison of the proportion of potentially inappropriate medication (PIM) six months after hospital discharge between the two treatment groups, the sample size calculation as well as the primary analysis should be based on the comparison of PIMs at month six. If other timepoint are considered as co-primary endpoints or as secondary endpoints should be decided and described. In the first case, a suitable multiple testing strategy should be considered. This strategy should then also be presented in detail! The likelihood ratio test across all time point does not prove the primary hypothesis. In addition, the sample size calculation should also focus on the contrast of the described model which address the comparison of PIMs at month six.

Moreover, some of the reads may be not familiar to the beta-regression approach. I would suggest explaining the concept with sufficient information to allow the analysis to be reproduced.

Item 20b; The authors stated that no additional analyses are planned (20b n.a.) could you please outline which data will be analyzed by t-, Wilcoxon-, Fisher-, Chi-square- or log-rank- (Kaplan-Meier estimates) test.

Moreover, some minor issues are to be noted;

The name of the intervention programme is used inconsistently, Please indicate the correct name MEDCORR or My Medication plan and please adjust throughout the protocol.

The sentence on line 102 “The enrolment, intervention and assessment” is incomplete, please complete

Figure 1 in line 126 should be labeled as Figure 2.

7. PLOS authors have the option to publish the peer review history of their article (what does this mean?). If published, this will include your full peer review and any attached files.

Reviewer #1: No

Reviewer #2: No

Reviewer #3: No

---

## [Author Response · Author response to Decision Letter 0]

7 Aug 2024

Dear Editor and Reviewers, 

Thank you for reviewing our manuscript and for your valuable suggestions and comments. Please find below our responses.

Reviewer #1: The authors designed a Randomized Controlled Trial to measure the Effect of a Medication Coordinator on the Quality of Patients’ Medication Treatment (MEDCOOR). However, the structure of the protocol does not fit with the guideline of the journal. 

- First, thank you for all the comments and suggested changes. We have now revised the manuscript so it fits within the guidelines of PLOSONE. 

Several breaks between paragraphs and unnecessary underlines hinder the quality of the paper. 

- We have revised the breaks between the paragraphs and removed unnecessary underlines.

The authors did not define or operationalize “Quality of Patients’ Medication Treatment”. Therefore, the topic and the objective that is going to be measured are not parallel. I would suggest a careful review of the protocol and its feasibility before implementation. Publication of the protocol in its current form is barely considered.

- Thank you for your feedback. We recognize that a clear definition was lacking. Hence, we have clarified the quality of patients’ medication treatment throughout the manuscript by adding a definition and how this is related to PIMs for this study protocol. See line 71-83 in the introduction section and line 191-202 in the method section. 

You might consider the following points to improve your work.

No Please reduce the number of keywords.

- Currently when re-submitting the manuscript, it is unfortunately not an option technically to reduce the number of keywords. 

- We have removed: Polypharmacy, Patient care management, Empowerment, Quality of life, and Treatment burden.

Lines 33-34“communication of communicating suggested medication amendments to the patients” should be revised.

- Thank you for the suggestion. We have implemented this in line 42-44. 

Line 190-192, Please describe the tool.

- We have included a brief description of the tool in the introduction section and 2.2.4. 

Line 203-205, no tool validation information is included.

- The tool, My Medication Plan, has been developed for this study, where face validation was a part of the process. It has not yet been validated as such. This study is a part of the validation and further validations and improvements of the tool, My Medication Plan, are planned in the future. 

Reviewer #2: Dear Authors

Thank you for your submission.

The comments will be available through journal editor.

Regards

- Thank you for your comments; they have been addressed throughout the revised manuscript based on the editor’s comment, which are found below.

Reviewer #3: The authors describe a protocol of a study which aims to evaluate the effect of a coordinating intervention on the quality of medication treatment. The primary objective of the planned study is the proportion of potentially inappropriate medication (PIM) six months after hospital discharge.

The description is detailed in many places; other points required by the SPIRIT guideline should be further elaborated.

- Thanks for this valuable input. We have adjusted the manuscript according to The SPIRIT Checklist where appropriate. 

- A new Supporting Information File - SPIRIT has been uploaded. 

Item 14; The sample size calculation in the manuscript is based on different assumptions and lead to a different number of patients to be includes (e.g. 93% power in the manuscript vs. 80% power in the study protocol resulting in n=70 patients per group in the manuscript or N=80 in the study protocol). Please explain deviations between the documents.

- Thank you for the comment. We apologize for the inconsistency, and we have adjusted the text according to advice by our statistician in the project group, hence the inclusion of 70 patients in the study protocol, but 80 in the original draft. 

The sample size calculation based on the assumption that the expected proportions of PIMS in the intervention group is higher as in the control group. Please explain why this assumption is reasonable.

- We suspect that the intervention will focus on PIMs, and therefore the number of PIMs in the intervention is higher, since our intervention is better at identifying them.

- Furthermore, we based our calculation on a study where this was the case. 

Item 15; Please add to the description of the clinic's resources which strategies will be used for achieving adequate participant enrolment to reach target sample size.

- This study is designed not to use any of the clinic resources, as the medication coordinator is an extra person. The intervention is intended to ease the workload of the clinic and support the clinic in relation to medication treatment. 

- A paragraph to describe this has been added at the end of section 2.1.1. 

Item 16a of the SPIRIT checklist suggests keeping details of any planned restriction (eg, blocking) in a separate document to conceal this knowledge to those who enroll participants or assign interventions. So the information from line 273 ff “A simple block randomization method …” should be transferred to a statistical analysis plan.

- As MS the main author of this article will assign, screen and recruit the patients, the information from line 273 and onwards cannot be concealed. 

Item 16b; Not knowing the block length alone cannot avoid predictability. Further measures to enable concealment should be considered and described in the protocol.

- We have added a section in the manuscript concerning blinding of the statistician in section 2.6.

Item 16c; Details of the Implementation of the allocation list should be provided. Who will generate the randomization list with REDCap, who will enrol patients and who will assign participants to interventions?

- The first author will conduct all the components of the study. Hence, MS will screen, recruit and assign participants to the intervention – or control group. The randomization is performed through REDCap, with blocks of varying sizes, which have been decided by the data manager. 

Item 20a; In general, the primary hypothesis, the sample size calculation and the statistical analysis should follow the same statistical approach. That means, if the primary objective is the comparison of the proportion of potentially inappropriate medication (PIM) six months after hospital discharge between the two treatment groups, the sample size calculation as well as the primary analysis should be based on the comparison of PIMs at month six. If other timepoint are considered as co-primary endpoints or as secondary endpoints should be decided and described. In the first case, a suitable multiple testing strategy should be considered. This strategy should then also be presented in detail! The likelihood ratio test across all time point does not prove the primary hypothesis. In addition, the sample size calculation should also focus on the contrast of the described model which address the comparison of PIMs at month six.

Moreover, some of the reads may be not familiar to the beta-regression approach. I would suggest explaining the concept with sufficient information to allow the analysis to be reproduced.

- As a consequence of your comment, We have decided that treatment effect should be evaluated across all time points, hence we evaluated that multiple testing is not necessary.

- We have added the following reference concerning beta regression: Ferrari SLP, Cribari-Neto F (2004). “Beta Regression for Modelling Rates and Proportions.” Journal of Applied Statistics, 31(7), 799–815. 

Item 20b; The authors stated that no additional analyses are planned (20b n.a.) could you please outline which data will be analyzed by t-, Wilcoxon-, Fisher-, Chi-square- or log-rank- (Kaplan-Meier estimates) test.

- Thanks for your comment. We have now emphasized in the manuscript that the baseline variables will be analyzed using the above mentioned statistical tests and no additional analysis as described in the protocol will be conducted.

Moreover, some minor issues are to be noted;

The name of the intervention programme is used inconsistently, Please indicate the correct name MEDCORR or My Medication plan and please adjust throughout the protocol.

- The name of the intervention is MEDCOOR. 

- My Medication Plan is a patient tool used during the medication review; hence it is a part of the intervention. 

- We are not able to identify any mention of the intervention as My Medication Plan, but please let us know if this is the case after the revision. 

The sentence on line 102 “The enrolment, intervention and assessment” is incomplete, please complete

- This sentence have been re-written, please see line 115-116. 

Figure 1 in line 126 should be labeled as Figure 2

- All figures, tables, and supporting information have been updated to ensure correct citation.

Editors Comments:

- The layout of the manuscript have now been XX to meet PLOS ONEs style requirements. 

- “Funding information” and “Financial Disclosure” match after revisiting both. 

- Unfortunately, some of the grants to have a grant number. 

 [(1) The University of Southern Denmark, one year salory to MS on 425.000 DKK and a grant of 10.000 DKK to be used on running cost. 

https://www.sdu.dk/en/forskning/phd/phd_skoler/phdskolensundhedsvidenskab

(2) The Region of Southrn Denmark gave one year salory to MS on 436.000 DKK 

https://regionsyddanmark.dk/fagfolk/forskning/region-syddanmarks-forskningspuljer/ph-d-puljen

(3) The University Hospital of Southern Denmark gave one year salory for MS on 572.000 DKK and a grant on 13.000 DKK to be used in running costs. 

https://sygehussonderjylland.dk/]. 

- The role of the funder have been stated at the end of the section describing the financial disclosure. 

- The funders had no role in the study. 

- The Role of Funder statement have been included in the revised Cover Letter.

- As for now the only data we have to share is the Do File used for the power calculation (S2 File), as this is the study protocol. 

- According to our ethics regulation, data of individual patients must not be exposed and available data will be anonymized.

- Do files used for the analysis will be made available. 

5. Ethics statement only appears at the end of the manuscript:

Your ethics statement should only appear in the Methods section of your manuscript. If your ethics statement is written in any section besides the Methods, please move it to the Methods section and delete it from any other section. Please ensure that your ethics statement is included in your manuscript, as the ethics statement entered into the online submission form will not be published alongside your manuscript. 

- The Ethics statement have now been included at the end of the Method Section, 2.8.

- Captions for Supporting Information files have been included at the end of the manuscript. 

- Any in-text citation have been updated.

---

## [Decision Letter · Decision Letter 1]

18 Sep 2024

PONE-D-24-18485R1Study Protocol: The Effect of a Medication Coordinator on the Quality of Patients’ Medication Treatment (MEDCOOR) – Randomized Controlled TrialPLOS ONE

Dear Dr. Schlunsen,

Thank you for submitting your manuscript to PLOS ONE. After careful consideration, we feel that it has merit but does not fully meet PLOS ONE’s publication criteria as it currently stands. Therefore, we invite you to submit a revised version of the manuscript that addresses the points raised during the review process.

We look forward to receiving your revised manuscript.

Kind regards,

Ramune Jacobsen

Academic Editor

PLOS ONE

Reviewers' comments:

Reviewer's Responses to Questions

**Comments to the Author**

1. Does the manuscript provide a valid rationale for the proposed study, with clearly identified and justified research questions?

Reviewer #2: Yes

Reviewer #3: Partly

2. Is the protocol technically sound and planned in a manner that will lead to a meaningful outcome and allow testing the stated hypotheses?

Reviewer #2: Yes

Reviewer #3: Partly

3. Is the methodology feasible and described in sufficient detail to allow the work to be replicable?

Reviewer #2: Yes

Reviewer #3: No

4. Have the authors described where all data underlying the findings will be made available when the study is complete?

Reviewer #2: Yes

Reviewer #3: Yes

5. Is the manuscript presented in an intelligible fashion and written in standard English?

Reviewer #2: Yes

Reviewer #3: Yes

6. Review Comments to the Author

You may also provide optional suggestions and comments to authors that they might find helpful in planning their study.

Reviewer #2: Dear Authors

Thank you for revising the manuscript.

I hope the best outcome from this study that will be helpful for healthcare.

Best wishes

Reviewer #3: Thank you for the revised manuscript and the explanations. For me, some points remain open;

The authors clarified that the main author of the manuscript will assign, screen and recruit the patients. This approach is associated with a high potential for bias. The more important it is to take appropriate measures to avoid influencing the allocation. In particular, appropriate concelament measures should be implemented. Please note that concealment should not be confused with blinding. Concealment describes the non-disclosure of the assignment until the assignment itself, while blinding means the non-disclosure of the assignment beyond the assignment. Blinding of the statistician is not suitable to ensure concealment.

The primary outcome is (re-)defined as (p. 9, line 206-207) “... the proportion of PIMs after hospital discharge. The PIMs are defined by four different screening tools.” Furthermore, the authors stated “As a consequence of your comment, We have decided that treatment effect should be evaluated across all time points, hence we evaluated that multiple testing is not necessary.” Nevertheless, it is still to be read (p- 12, line 310-311) that “To avoid inflating the type I error rate in the z-test, we will adjust the z-test for multiple tests.” Please clarify, which timepoint are actually included (In figure 1 it is indicated that PIMs are evaluated at t1, t2 and t6 – are this baseline, 3 and 6 months?), how the primary outcome is precisely defined (as change from baseline to any timepoint after discharge or value at any timepoint after discharge), and if adjusting the significance level for multiple testing will be an issue or not. In this context, it is equally important to make clear how the four different screening tools for defining PIMs will be applied and combined.

In response to my question as to what the various tests mentioned are used for, the authors stated “We have now emphasized in the manuscript that the baseline variables will be analyzed using the above mentioned statistical tests and no additional analysis as described in the protocol will be conducted. “One of the recommendations of the CONSORT statement is that a table is presented showing baseline demographic and clinical characteristics for each group. Importantly, according to this recommendation in the CONSORT statement, “significance testing of baseline differences in randomized controlled trials (RCTs) should not be performed, because it is superfluous and can mislead investigators and their readers” (c.f. de Boer et al. International Journal of Behavioral Nutrition and Physical Activity (2015) 12:4).

7. PLOS authors have the option to publish the peer review history of their article (what does this mean?). If published, this will include your full peer review and any attached files.

Reviewer #2: No

Reviewer #3: No

---

## [Author Response · Author response to Decision Letter 1]

11 Oct 2024

Dear Reviewers, 

Thank you for your suggestions and comments. Please see our answers. 

Reviewer #2: Dear Authors

Thank you for revising the manuscript.

I hope the best outcome from this study that will be helpful for healthcare.

Best wishes

- Thank you for reading the manuscript once again, we appreciate it. 

Reviewer #3: Thank you for the revised manuscript and the explanations. For me, some points remain open;

The authors clarified that the main author of the manuscript will assign, screen and recruit the patients. This approach is associated with a high potential for bias. The more important it is to take appropriate measures to avoid influencing the allocation. In particular, appropriate concealment measures should be implemented. Please note that concealment should not be confused with blinding. Concealment describes the non-disclosure of the assignment until the assignment itself, while blinding means the non-disclosure of the assignment beyond the assignment. Blinding of the statistician is not suitable to ensure conceal-ment.

- Thank you for your comment. We agree that it is a concern that blinding is not possible; hence, there is a risk of bias. 

- The allocation is random and concealed as the allocation is performed after the inclusion of the pa-tient. The first author cannot influence the randomization. 

- This study represents the best option; as other blinding methods were not feasible due to the inter-vention in itself. 

The primary outcome is (re-)defined as (p. 9, line 206-207) “... the proportion of PIMs after hospital dis-charge. The PIMs are defined by four different screening tools.” Furthermore, the authors stated “As a consequence of your comment, We have decided that treatment effect should be evaluated across all time points, hence we evaluated that multiple testing is not necessary.” Nevertheless, it is still to be read (p- 12, line 310-311) that “To avoid inflating the type I error rate in the z-test, we will adjust the z-test for multi-ple tests.” Please clarify, which timepoint are actually included (In figure 1 it is indicated that PIMs are evaluated at t1, t2 and t6 – are this baseline, 3 and 6 months?), how the primary outcome is precisely de-fined (as change from baseline to any timepoint after discharge or value at any timepoint after discharge), and if adjusting the significance level for multiple testing will be an issue or not. In this context, it is equally important to make clear how the four different screening tools for defining PIMs will be applied and combined.

- Thank you for your comment. The primary analysis is a global test hence no adjustment for multi-ple testing will be utilized in that analysis. Denote that the analysis where we adjust for multiple testing is not the primary analysis but a secondary one, which only will be conducted if the global test is statistically significant. We have added on line 307 that the measurement is taken on base-line, at hospital discharge and 6 month after hospital discharge.

- The definition of PIMs is the amount of potentially inappropriate medication of the total medica-tions of the given patient. A potential inappropriate medication is defined if at least one of the four tools indicates it. We have added/revised? the following formulation on line 207: “, hence if at least one tool indicates that a medication as a PIM, we will define the medication as a PIM” 

In response to my question as to what the various tests mentioned are used for, the authors stated “We have now emphasized in the manuscript that the baseline variables will be analyzed using the above men-tioned statistical tests and no additional analysis as described in the protocol will be conducted. “One of the recommendations of the CONSORT statement is that a table is presented showing baseline demo-graphic and clinical characteristics for each group. Importantly, according to this recommendation in the CONSORT statement, “significance testing of baseline differences in randomized controlled trials (RCTs) should not be performed, because it is superfluous and can mislead investigators and their readers” (c.f. de Boer et al. International Journal of Behavioral Nutrition and Physical Activity (2015) 12:4).

- Thank you for you reflection. We agree that the usage of p-values in descriptive tables is inappro-priate however in order to not restrict ourselves from a given journal, we will include p-values unless we are told otherwise or the author guidelines for the given journal states it clearly.

---

## [Decision Letter · Decision Letter 2]

5 Nov 2024

Study Protocol: The Effect of a Medication Coordinator on the Quality of Patients’ Medication Treatment (MEDCOOR) – Randomized Controlled Trial

PONE-D-24-18485R2

Dear Dr. Schlunsen,

We’re pleased to inform you that your manuscript has been judged scientifically suitable for publication and will be formally accepted for publication once it meets all outstanding technical requirements.

Kind regards,

Ramune Jacobsen

Academic Editor

PLOS ONE

Reviewers' comments:

Reviewer's Responses to Questions

**Comments to the Author**

1. Does the manuscript provide a valid rationale for the proposed study, with clearly identified and justified research questions?

Reviewer #3: Yes

2. Is the protocol technically sound and planned in a manner that will lead to a meaningful outcome and allow testing the stated hypotheses?

Reviewer #3: Yes

3. Is the methodology feasible and described in sufficient detail to allow the work to be replicable?

Reviewer #3: Yes

4. Have the authors described where all data underlying the findings will be made available when the study is complete?

Reviewer #3: Yes

5. Is the manuscript presented in an intelligible fashion and written in standard English?

Reviewer #3: Yes

6. Review Comments to the Author

You may also provide optional suggestions and comments to authors that they might find helpful in planning their study.

Reviewer #3: All my comments from the first and second round of the review have been sufficiently addressed, thank you.

7. PLOS authors have the option to publish the peer review history of their article (what does this mean?). If published, this will include your full peer review and any attached files.

Reviewer #3: No

---

## [Editor Report · Acceptance letter]

15 Nov 2024

PONE-D-24-18485R2 

PLOS ONE

Dear Dr. Schlunsen, 

I'm pleased to inform you that your manuscript has been deemed suitable for publication in PLOS ONE. Congratulations! Your manuscript is now being handed over to our production team.

Kind regards, 

on behalf of

Dr. Ramune Jacobsen 

Academic Editor

PLOS ONE